# Interatomic Potential for InP

**DOI:** 10.3390/ma15144960

**Published:** 2022-07-16

**Authors:** Dariusz Chrobak, Anna Majtyka-Piłat, Grzegorz Ziółkowski, Artur Chrobak

**Affiliations:** 1Institute of Materials Engenering, Faculty of Science and Technology, University of Silesia in Katowice, 75 Pułku Piechoty 1A, 41-500 Chorzow, Poland; anna.majtyka@us.edu.pl; 2Institute of Physics, University of Silesia in Katowice, 75 Pułku Piechoty 1A, 41-500 Chorzów, Poland; grzegorz.ziolkowski@us.edu.pl (G.Z.); artur.chrobak@us.edu.pl (A.C.)

**Keywords:** indium phosphide, molecular dynamics simulations, interatomic potential, phase transformation, native point defects

## Abstract

Classical modeling of structural phenomena occurring in InP crystal, for example plastic deformation caused by contact force, requires an interatomic interaction potential that correctly describes not only the elastic properties of indium phosphide but also the pressure-induced reversible phase transition B3↔B1. In this article, a new parametrization of the analytical bond-order potential has been developed for InP. The potential reproduces fundamental physical properties (lattice parameters, cohesive energy, stiffness coefficients) of the B3 and B1 phases in good agreement with first-principles calculations. The proposed interaction model describes the reversibility of the pressure-induced B3↔B1 phase transition as well as the formation of native point defects in the B3 phase.

## 1. Introduction

Indium phosphide attracts significant interest among all semiconductors due to its wide applications in electronics and photonics, for example, in applications in high-speed integrated circuits [1], photonic devices [2,3], photovoltaic [4], and high-efficiency solar cells [5]. The high absorption coefficient allows the InP to be an essential constituent also of many smart devices as quantum-dot light emitters [6,7], nanowires lasers [8], and nanowire photodetectors [9].

A broad range of applications of InP stipulates growing interest in theoretical modeling of its atomic structure under various conditions. One can consider a quantum-mechanical (first-principles) calculations based on density functional theory (DFT) that are now routinely used to investigate, e.g., high-pressure phase transitions [10], native point defects [11], electronic [12,13], and elastic properties [14]. On the other side, although this method is highly transferable and gives good results, it uses a significant amount of computational resources, which limits simulations to small systems consisting of no more than a few hundred atoms. Thus, the development of empirical potentials for large-scale computer simulation based on classical molecular dynamics (MD) is of great interest, and several potentials have been proposed so far.

Powell et al. [15] developed a Tersoff-like potential, which allowed them to predict the elastic and vibrational properties of the B3 (zinc blende) phase of InP. Han et al. [16] offered interatomic potentials based on the Stillinger–Weber [17] functional form (supplemented by a long-range Coulomb interaction) for vibrational properties calculations. Another long-distance interactions scheme has been employed by Branico et al. [18,19] to reproduce the lattice constant, stiffness coefficients, the course of the B3→B1 phase transformation, surface energies, and the generalized stacking fault energy curve. Although the Branicio potential is well suited for the low-pressure B3 phase, it exhibits limited applicability in case of native point defects simulations and correct description of the reversible B3↔B1 phase transformation. Furthermore, Nayir et al. [20] used the Branicio potential to investigate a size-dependence of the heat capacity of InP nanostructures. Interest in MD simulations using potentials designed for InP has recently been confirmed in studies of the thermal conductivity at InGaAs/InP interfaces [21] and the mechanical properties of indium phosphide nanowires [22].

In the present work, we study the B3↔B1 phase transition in indium phosphide using the classical molecular dynamics (MD) method and a new interatomic potential based on the bond-order (BO) model of multi-body interactions [23,24,25,26]. In addition to the structural and elastic properties of the B3 and B1 phases, the designed potential correctly models the energy of formation of various native point defects in the B3 phase.

## 2. Materials and Methods

### 2.1. First-Principles Calculations

First-principles calculations were based on the density functional theory (DFT). We used the *Quantum-Espresso* software [27] with ultrasoft pseudopotentials for In and P elements (https://pseudopotentials.quantum-espresso.org/, accessed on 10 July 2022). The LDA-PZ [28] and GGA-PBE [29] functionals were selected to approximate the exchange-correlation energy. The calculations assumed a non-magnetic ground state. In order to achieve high computational stability, each electronic state was expressed by a series of plane waves limited by the kinetic energy cut-off 160 Ry. At the same time, the first Brillouin zone was sampled by 16×16×16 Monkhorst-Pack mesh [30]. The DFT calculation was used to complement the set of physical properties of various phases of InP, In, and P. In order to estimate the cohesive energies of considered crystal lattices, we also calculated the spin-polarized ground state energy of free In and P atoms.

### 2.2. Molecular Dynamics Simulations

Classical molecular dynamics simulations (MD) were performed using the LAMMPS software [31] for modeling a course of the pressure-induced B3←B1 phase transformation in an InP crystal. For that goal, a supercell containing 16×16×16 unit cells of the B3 phase was employed. The Velocity–Verlet time integration algorithm, with an increment Δt=2 fs, was used through the simulations, while the NPT ensemble controlled the indicated thermodynamic variables. Before running simulations, the system was relaxed (10,000 Δt) to reach equilibrium at 300 K and 0 GPa. The pressure acting on the InP crystal was then altered from 0 to 18 GPa and back to 0 GPa (total 2,000,000 Δt). Structural changes were recorded every 10,000 time steps. The OVITO software [32] was used for the visualization of structural changes. Moreover, MD simulations supplemented by the method described in work by Wang et al. [33] were applied for calculations of the bulk modulus *B* and the elastic constants $c_{11}$, $c_{12}$, $c_{44}$ of considered cubic crystals.

### 2.3. The Atomic Bond Order Potential

Selecting the model for interatomic interaction in InP crystal, we focused on the transferability of the analytical bond-order (BO) potential, the feature which was previously demonstrated for, e.g., SiC [26], FePt [24], and GaAs [25] materials.

The following formula expresses the potential energy of *N* interacting atoms:(1)$$U=\frac{1}{2}\sum_{i}\sum_{j\neq i}V_{ij}$$
$$V_{ij}=f_{c}\left(r_{ij}\right)\left[V_{R}(r_{ij}+b_{ij}V_{A}\left(r_{ij}\right)\right]$$
where the cut-off function $f_{c}$ defines the range of interactions:$$f_{c}\left(r_{ij}\right)=\left\{\begin{array}{cc} 1 & \mathrm{if}\;r_{ij}<R+D\\ \frac{1}{2}-\frac{1}{2}\sin\left(\frac{\pi }{2}\frac{r_{ij}-R}{D}\right) & \mathrm{if}\;R-D\geq r_{ij}\leq R+D\\ 0 & \mathrm{if}\;r_{ij}>R-D \end{array}\right.$$
The two-body repulsive $V_{R}$ and attractive $V_{A}$ terms depend on four parameters:(2)$$V_{R}\left(r_{ij}\right)=\frac{D_{0}}{S-1}\exp\left(-\beta \sqrt{2S}\left(r_{ij}-r_{0}\right)\right)$$
(3)$$V_{A}\left(r_{ij}\right)=-\frac{SD_{0}}{S-1}\exp\left(-\beta \sqrt{2/S}\left(r_{ij}-r_{0}\right)\right)$$
while the bond-order term $b_{ij}$ tunes the strength of two-body interactions and depends on another eight parameters:(4)$$b_{ij}={\left[1+\beta_{2}^{n}\zeta_{ij}^{n}\right]}^{-\frac{1}{2n}}$$
(5)$$\zeta_{ij}=\sum_{k\neq ij}f_{c}\left(r_{ij}\right)g\left(\theta_{ijk}\right)\exp\left[\lambda_{3}^{m}{\left(r_{ij}-r_{ik}\right)}^{m}\right]$$
(6)$$g\left(\theta_{ijk}\right)=\gamma_{ijk}\left(1+\frac{c^{2}}{d^{2}}-\frac{c^{2}}{d^{2}+{\left(h+\cos\theta_{ijk}\right)}^{2}}\right)$$
where $\theta_{ijk}$ stands for the bond-angle describing a space arrangement of *i*, *j*, and *k* atoms.

The parameters appearing in the above equations should be selected in such a way as to ensure the best compatibility of the MD simulation results with the reference data. Some parameters have a direct interpretation. The value of $D_{0}$ (multiplied by minus one) is the binding energy of the dimer characterized by a distance between atoms equal to $r_{0}$. *R* determines the range of interactions, and *D* is the width of a region where interactions fall to zero in a sinusoidal manner.

### 2.4. Fitting Procedure

In order to find a parametrization of the BO potential which adequately reproduces the desired physical properties of InP crystal, we employed *Atomicrex* code [34], which minimizes the objective function of the general form:(7)$$F\left\{D_{0},S,\ldots \right\}=\sum_{i=1}^{N}w_{i}{\left[A_{i}^{ref}-A_{i}\left\{D_{0},S,\ldots \right\}\right]}^{2}$$
where: *N*—the number of crystal properties used in the fitting procedure, $w_{i}$—the i-th weight ($\sum w_{i}=1$), $A_{i}^{ref}$—the reference value of the i-th crystal property, $A_{i}\left\{D_{0},S,\ldots \right\}$—the value of the i-th property calculated with the BO potential. In order to find the minimum of the objective function, we used the Nelder–Mead simplex algorithm.

The minimization process must start from a carefully selected set of potential parameters. In the present work, we follow the method used by Albe et al. [24,25], who employed the Pauling equation to find the initial guess of two-body parameters:(8)$$E_{b}=-D_{0}\exp\left[-\beta \sqrt{2S}\left(r_{b}-r_{0}\right)\right]$$
where $E_{b}$—the energy per bond, $r_{b}$—the bond distance. The initial set of three-body parameters for In-P-P, In-In-In, and P-P-P interactions was adopted from the article by Powell et al. [15].

## 3. Results and Discussion

### 3.1. Parametrization of the BO Potential

Table 1 presents structural, elastic, and energetic properties of the B3 and B1 phases of indium phosphide calculated within the frame of the first-principles method. A comparison with attached reference data convinces that the LDA-PZ approximation of the exchange-correlation functional is a good choice for the lattice constant, the bulk modulus, and stiffness coefficients (*a*, *B*, $c_{ij}$). In agreement with previous works [35,36], we observed that the GGA-PBE approximation of the exchange–correlation functional is more suitable for cohesive energy calculations. Therefore, the InP crystal we intend to model has properties with numerical values highlighted in Table 1. We also examined the properties of dimer, diamond, RS-like (two FCC lattices shifted by the vector ($\frac{1}{2}\frac{1}{2}\frac{1}{2}$)), and FCC structures of In and P crystals. However, in this case, focusing on the essential task of building the interaction potential for indium phosphide, we limited ourselves only to lattice constants, cohesive energies, and bulk moduli. As shown in Table 1, we also calculated the cohesive energy per bond $E_{b}$ and the bond length $r_{b}$, as they are necessary for the calculation (via Pauling equation) of the initial values of two-body parameters of the BO potential.

Table 2 presents the formation energy of various native point defects in the B3 phase of InP. The data come from an article by Mishra et al. [11], who used the first-principles method for investigations of vacancies ($V_{In}$, $V_{P}$), antisites ($In_{P}$, $P_{In}$) and interstitials ($In_{i}$, $P_{i}$). The $In_{i}$ point defect can be imagined as an additional In atom located in the center of the B3 unit cell and surrounded by four other In atoms. The $P_{i}$ point defect is characterized by two P atoms located inside the In-tetrahedron and aligned in the <110> direction (so-called <110>-split defect).

Calculations of the formation energy of native point defects are based on the fundamental relationship between the chemical potentials:(9)$$\mu_{In}+\mu_{P}=\mu_{InP}=E\left(In_{N}P_{N}\right)/N$$
and equations directly defining the formation energies:(10)$$\begin{array}{cc} E(V_{P}) & =E\left(V_{P},In_{N}P_{N-1}\right)-\left(E\left(In_{N}P_{N}\right)-\mu_{P}\right)\\ E(V_{In}) & =E\left(V_{In},In_{N-1}P_{N}\right)-\left(E\left(In_{N}P_{N}\right)-\mu_{In}\right)\\ E(P_{In}) & =E\left(P_{In},In_{N-1}P_{N+1}\right)-\left(E\left(In_{N}P_{N}\right)-\mu_{In}+\mu_{P}\right)\\ E(In_{P}) & =E\left(In_{P},In_{N+1}P_{N-1}\right)-\left(E\left(In_{N}P_{N}\right)-\mu_{P}+\mu_{In}\right)\\ E(P_{i}) & =E\left(P_{i},In_{N}P_{N+1}\right)-\left(E\left(In_{N}P_{N}\right)+\mu_{P}\right)\\ E(In_{i}) & =E\left(In_{i},In_{N+}P_{N}\right)-\left(E\left(In_{N}P_{N}\right)+\mu_{In}\right) \end{array}$$
where, for example, $E(V_{P})$—the formation energy of P-vacancy, $E(V_{P},In_{N}P_{N-1})$—the energy of a relaxed crystal containing P-vacancy, $E(In_{N}P_{N})$—the energy of a relaxed pure crystal composed of *N* formula units, and $\mu_{P}$—the chemical potential of P.

The application of the above equations presents a particular difficulty, since the knowledge of $\mu_{In}$ and $\mu_{P}$ chemical potentials is needed. In order to overcome this problem, we considered the InP crystal under non-stoichiometric In-rich and P-rich conditions. The method, for a given type of non-stoichiometry, assumes a constitutional defect with the lowest formation energy among all other point defects. Then, the state of the crystal containing such a point defect is considered a ground state. As a consequence, the formation energy of a constitutional point defect has to vanish. Furthermore, by zeroing one of the Equations (10) corresponding to a constitutional point defect and using formula (9), we obtain the possibility of calculating $\mu_{In}$ and $\mu_{P}$ chemical potentials. Then, the computation of the formation energy of other native point defects is uncomplicated. Mishra et al. [11] showed that $In_{P}$ and $P_{In}$ antisites play a role of constitutional native point defects for In-rich and P-rich conditions, respectively.

Having defined the set of physical properties of the InP, In, and P crystals, we used the Atomicrex software to find a satisfactory parametrization of the BO potential. The initial values of two-body potential parameters were obtained by application of the Pauling equation to $E_{b}\left(r_{b}\right)$ data collected for In-In, P-P, and In-P interactions. The three-body parameters for all interactions were adopted from the article by Powell et al. [15], except for substitution $\beta_{2}=1$ and m=1, which simplifies the fitting procedure. The parameters of the cut-off function (*R* and *D*) were selected to account for the interactions between nearest neighbors only. Thus, the BO potential effectively depends on ten parameters.

Initially, the fitting procedure was applied to the InP, In, and P crystals separately. It resulted in three independent sets of the BO potential parameters. Then, we constructed the potential file appropriate for LAMMPS simulations using the method described in the software documentation. At this stage, the potential reproduced the basic properties of InP, In, and P crystals included in Table 1. Unfortunately, the values of the native point defects formation energies were completely wrong. Attempts to solve this problem have shown that the reliable modeling of formation energies of native point defects and the correct description of physical properties of the B3 and B1 phases of InP is possible; however, it takes place under slight deviation of two-body parameters of In-In and P-P interactions. The final parametrization of the BO potential is presented in Table 3. The satisfactory description of the target InP crystal and the native point defects in its B3 structure has cost the loss of the opportunity of modeling the considered In and P phases.

### 3.2. B3→B1 Phase Transformation

The B3 phase of InP transforms into the B1 phase at pressure approx. 10.8GPa [41,42]. The high-pressure phase of InP exhibits metallic properties which contrast the semiconducting nature of the initial B3 phase. The doping of InP crystal affects the pressure of the B3→B1 phase transformation [10,43,44]. It was shown that in the InP crystal doped by sulfur, the phase transition occurs in the pressure range from 10.4 to 13.3 GPa [41]. Similarly, investigations performed on crystals doped by iron showed that the transformation starts at 9.5 GPa and finishes at 13.1 GPa [42]. The B3→B1 phase transformation is reversible [41,42].

In order to provide a rough estimation of the B3→B1 transformation pressure, we considered two systems composed of 4×4×4 unit cells of B3 and B1 phases. Then, using LAMMPS software, we calculated the pressure dependence of the enthalpy *H* using both the BO and Branicio [18] potentials. A closer inspection of the obtained results (Figure 1a) shows that the curves representing the B3 and B1 phases cross at the point where abscissa equals 13.1 GPa and 10.3 GPa for the BO and Branicio potentials, respectively. One can see that the Branicio potential gives the value of transformation pressure very close to experimental results. A more demanding test is presented below.

The pressure-induced B3↔B1 phase transformation at the temperature of 300 K was simulated using the procedure described in the Method section. In order to bring the simulations closer to real conditions, the perfect InP crystal (16×16×16 B3 unit cells) was perturbed by introducing vacancies (0.001 percent). Then, the system was compressed from 0 to 18 GPa and decompressed to 0 GPa. The full path of pressure change was linearly completed during 2,000,000 time steps. Figure 1b shows the pressure *p* dependence of the crystal volume *V* obtained for both the BO and Branicio potentials. One can see that the phase transformation, indicated by the sudden drop in the volume, starts at 13.9 GPa and 14.7 GPa for the BO and Branicio potentials, respectively. The simulations (not shown) performed on a perfect crystal gave higher pressures: 16.1 GPa for the present and 22.0 GPa for Branicio potential.

Interestingly, the simulations with the BO potential exhibit lower pressure of the B3↔B1 transition. This effect may be due to the fundamental difference between the mathematical models of the BO and Branicio potential. Namely, the latter is a long-range potential. A larger interaction range can be expected to stabilize the B3 phase to a greater extent than in the case of short-range interactions. Thus, the width of the hysteresis of the B3↔B1 transformation modeled by the Branicio potential should be greater than in the case of the BO potential (see Figure 1b). It is worth noting that the similar effect of long-range interactions on the hysteresis loop width was also registered in Monte Carlo simulations of magnetic properties of the Ising-like lattice [45].

Figure 2 confirms that the B3→B1 indeed occurred. The panel (a) presents a 20Å thick slice of the modeled crystal cut perpendicularly to the [110] direction. One can see that at the end of compression, at 18 GPa, the crystal is composed of grains of a high-pressure phase. Direct inspection of its unit cell confirms the B1 structure (Figure 2b). The average length of the presented B3 unit cell edges equals 5.171Å.

A first-principles study focused on the mechanism of the B3→B1 phase transition in ZnS crystal was published by Catti [46]. The author considered the orthorhombic and rhombohedral pathways. The first mechanism is characterized by an intermediate state of Pmm2 symmetry, whereas the second occurs by the state of R3m symmetry. In the orthorhombic transition path, Zn (or S) atoms are displaced to the midpoint between S (or Zn) atoms at the tetrahedron corners. It was shown that the orthorhombic transition path is more favorable.

An inspection of atomic displacements during the phase transformation in InP crystal modeled by BO potential allowed us to conclude that the B3→B1 phase transformation employs the orthorhombic mechanism. Indeed, Figure 3a shows a group of atoms in the B3 phase, viewed along the [0–11] direction. Figure 3b displays the same atomic system slightly rotated and with several atoms colored for better visualization of their displacements. For example, during the phase transformation, the P atom (yellow, index 1) and its two nearest neighbors In atoms (magenta, index 2 and 3) move in such a way that they form a linear In–P–In arrangement. Other atoms behave similarly, and finally, they occupy positions appropriate for the B1 phase unit cell.

The V(p) curves (Figure 1b) obtained for both the BO and Branicio potentials reveal a significant difference. Indeed, the transformation hysteresis loop modeled by the Branicio potential is open, which contrasts the closed hysteresis simulated with the BO potential. The reversibility of the B3→B1 transformation simulated with the BO interaction model is confirmed in Figure 4, which shows radial distribution function (RDF) calculated for three states of the crystal: initial (0 GPa), under maximal pressure of 18 GPa, and final, after decompression to 0 Gpa. For the initial state, interatomic distances are distributed around 2.530Å, 4.124Å, and 4.842Å, which characterizes the B3 phase with the lattice constant of 5.838Å. After the phase transformation, at the pressure of 18 GPa, the RDF calculated with the BO potential exhibits major peaks around 2.444Å, 3.661Å, 4.406Å, and 4.778Å. The first peak reflects the distance between In–P nearest neighbors, and the second peak reflects the origins from the closest In–In and P–P atoms. The last two peaks (red circles) are related to the distance between the In (or P) atoms in the corner and the P (or In) atoms located at the center of the B1 unit cell. The splitting occurs because of a slight relative shift (∼0.3Å) of the In and P sublattices in the ${\left[110\right]}_{B1}$ direction. Further analysis showed that the mean value of the B1 lattice constant at 18 GPa and 300 K equals 5.123Å. The radial distribution function calculated at 18 GPa with the Branicio potential exhibits major peaks around 2.571Å, 3.415Å, 3.943Å, and 4.747Å. The peaks at 3.415Å and 3.943Å (green circles) result from the slight rhombohedral distortion of the unit cell base. The mean lattice constant of the B3 phase equals 5.183Å.

After decompression, the crystal modeled by the BO potential returned to the initial B3 structure. In contrast, the decompression of the system modeled by the Branicio potential left the crystal in the distorted B1 phase. The distortion’s nature is represented by the first two RDF peaks (blue circles), which are related to separated ranges of the In–P nearest neighbors distances.

## 4. Conclusions

This work presents the development of the analytical bond-order potential designed for InP crystal. The interaction model accurately describes the lattice constant, the cohesive energy, and the stiffness coefficients of the B3 and B1 phases. Our potential also reproduces the formation energy of native point defects in the B3 phase under In-rich and P-rich conditions. However, due to the short-range nature of interactions, one can expect that the potential exhibits limited capability for a description of vibrational properties of InP. The proposed potential also fails in modeling indium phosphide’s melting temperature because this property was not included in reference data for the fitting procedure.

The molecular dynamics simulations showed the capability of the BO potential for modeling the reversible B3↔B1 phase transformation. This feature of the BO potential contrasts the results obtained with the Branicio potential that cannot model the reverse B1→B3 transition. Among the two pathways of atomic displacements considered in this paper that lead from the B3 to B1 structure, our BO potential realizes the orthorhombic path. The potential, we proposed, also has such an advantage over so far known potentials that it gives a reliable description of the formation energies of the native point defects.

While searching for the application of the new InP potential, attention can be paid to simulations of deformation phenomena in the nanoscale, which requires a potential that models elastic properties of the B3 and B1 phases and accurately describes the course of the B3→B1 transformation. Moreover, these simulations could be performed on the system containing native point defects.

## Figures and Tables

**Figure 1 materials-15-04960-f001:**
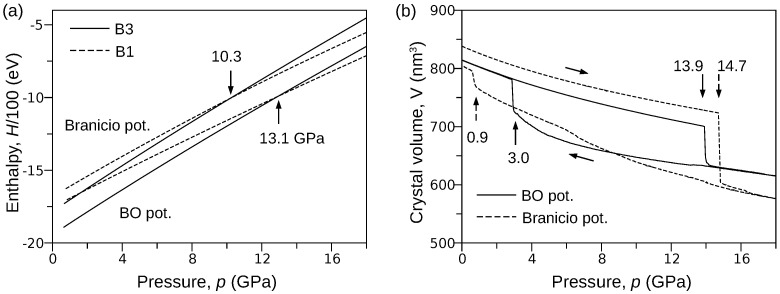
Results of MD simulations performed with the BO and Branicio potentials. (**a**) The relationship between the enthalpy *H* and the pressure *p* calculated for the B3 and B1 phases of InP crystal (4×4×4 unit cells). (**b**) Compression of the InP crystal (16×16×16 unit cells). The hysteresis of the B3↔B1 transformation represented by the V(p) relationship (BO potential) reveals the reversibility of the phase transition.

**Figure 2 materials-15-04960-f002:**
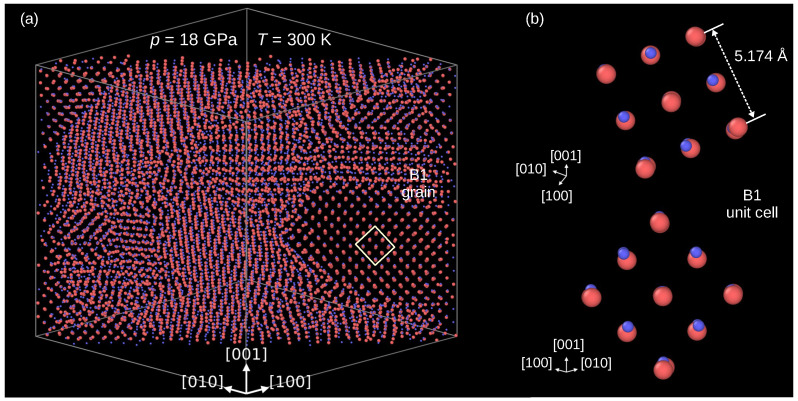
The InP crystal after the B3→B1 phase transformation. (**a**) Multi-grain structure of the the modeled crystal. (**b**) Details of atomic arrangements in the B1 phase. Atoms: In—coral, P—blue.

**Figure 3 materials-15-04960-f003:**
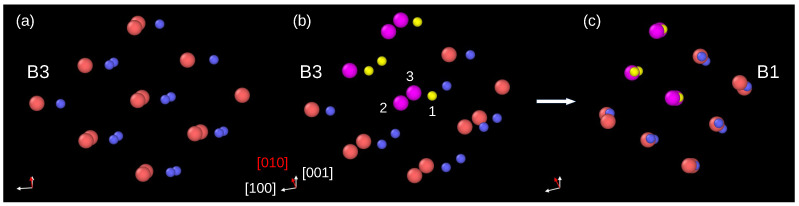
Details of the B3→B1 phase transformation. (**a**) Group of atoms in the B3 phase right before the phase transformation viewed along the [0–11] direction. (**b**) The same group of atoms; however, the direction of view was slightly changed for a better presentation of further atom displacements. Several atoms are colored: the P atoms—yellow, In atoms—magenta. (**c**) After the phase transformation, the atoms form the B1 unit cell. The orthorhombic pathway of the phase transformation can be traced using atoms marked as 1, 2, and 3. The atom 1 (P) displaces into the site between its nearest neighbors 2 (In) and 3 (In).

**Figure 4 materials-15-04960-f004:**
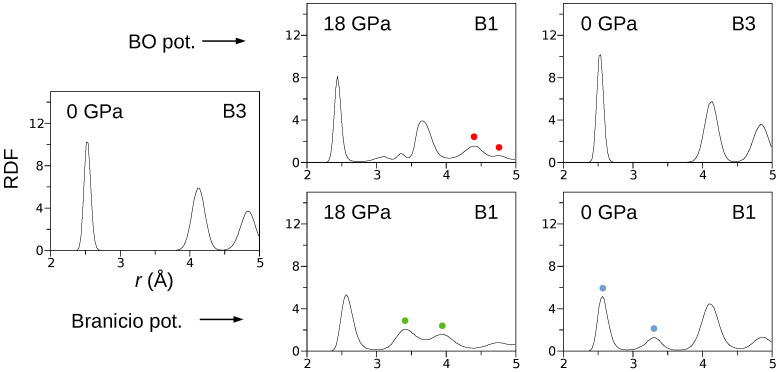
The radial distribution functions calculated B3↔B1 phase transformations. The presented data reveal the complete structural reversibility of the phase transformation modeled by the BO potential.

**Table 1 materials-15-04960-t001:** The structural, elastic, and cohesive properties of various crystalline phases of InP, In and P: the lattice constant *a* (Å), the bond length $r_{b}$ (Å), the cohesive energy per formula unit $E_{coh}$ (eV), the cohesive energy per bond $E_{b}$ (eV), the bulk modulus *B* (GPa), and stiffness coefficients $c_{ij}$ (GPa). The reference data used for fitting are highlighted.

	DFT				Other			
	LDA		GGA		Data	Branicio		This Work
**InP B3**								
*a*	**5.829**		5.965		5.869 [37]	5.869		5.831
$E_{coh}$	7.99		**6.24**		6.72 [38]	6.97		7.59
$r_{b}$	**2.524**		2.583		2.541	2.541		2.525
$E_{b}$	2.00		**1.56**		1.68	1.74		1.90
*B*	**71.1**		59.1		71.1 [37]	71.5		69.1
$c_{11}$	**100.4**		86.5		101.1 [37]	101.4		94.3
$c_{12}$	**56.4**		45.4		56.1 [37]	56.5		56.5
$c_{44}$	**45.3**		41.1		45.6 [37]	37.7		44.1
**InP B1**								
*a*	**5.418**		5.546		5.42 [39]	5.643		5.391
$E_{coh}$	7.64		**5.75**			6.57		6.85
$r_{b}$	**2.709**		2.773			2.822		2.696
$E_{b}$	1.27		**0.96**			1.64		1.41
*B*	**89.9**		73.4		88.6 [39]	49.5		90.0
$c_{11}$	**182.5**		146.9		246.6 [40]	71.5		186.2
$c_{12}$	**43.6**		36.6		48.8 [40]	38.5		41.9
$c_{44}$	**31.8**		27.7		30.7 [40]	20.4		32.6
In								
	dimer		diamond		RS like		FCC	
	LDA	GGA	LDA	GGA	LDA	GGA	LDA	GGA
*a*			**6.463**	6.737	**5.940**	6.186	**4.616**	4.792
$E_{coh}$			2.67	**2.00**	2.97	**2.24**	3.04	**2.33**
$r_{b}$	**2.930**	3.092	**2.799**	2.917	**2.970**	3.093	**3.264**	3.389
$E_{b}$	1.64	**1.42**	1.34	**1.00**	0.99	**0.75**	0.51	**0.39**
*B*			**32.3**	21.4	**45.0**	30.9	**50.0**	35.8
**P**								
*a*			**5.337**	5.495	**4.822**	4.889	**3.811**	3.879
$E_{coh}$			4.00	**2.90**	4.55	**3.33**	3.82	**2.59**
$r_{b}$	**1.887**	1.905	**2.311**	2.379	**2.411**	2.445	**2.695**	2.743
$E_{b}$	6.44	**5.16**	2.00	**1.45**	1.52	**1.11**	0.64	**0.43**
*B*			**51.4**	43.2	**125.9**	109.1	**107.8**	90.2

**Table 2 materials-15-04960-t002:** The formation energy $E_{f}$ (eV) of native point defects in B3 phase of the InP crystal.

B3 Native Point Defects, $E_{f}$			
	**Mishra (DFT)**	**Branicio**	**This Work**
**In** rich			
$\mu_{In}$, $\mu_{P}$	−1.52, −6.94	−0.03, −6.94	−1.66, −5.93
$V_{In}$	5.26	6.34	5.21
$V_{P}$	0.98	−1.03	0.93
$In_{P}$	0.0	0.0	0
$P_{In}$	4.88	16.07	4.88
$In_{i}$	1.62	−0.55	1.66
$P_{i}$	4.19	8.41	5.81
**P** rich			
$\mu_{In}$, $\mu_{P}$	−3.96, −4.50	−8.06, 1.09	−4.10, −3.49
$V_{In}$	2.83	−1.69	2.77
$V_{P}$	3.42	7.00	3.37
$In_{P}$	4.88	16.07	4.88
$P_{In}$	0.0	0.0	0
$In_{i}$	4.06	7.48	4.10
$P_{i}$	1.76	0.37	3.37

**Table 3 materials-15-04960-t003:** The BO potential parameters (see Equations (1)–(6)) optimized for the InP crystal.

$D_{0}$	*S*	β	$r_{0}$	*n*	γ	*c*	*d*	*h*	$\lambda_{3}$
(eV)		(1/Å)	(Å)						(1/Å)
**In-P**									
2.039682	2.218147	1.410176	2.500450	4.768664	0.209249	0.686344	0.411612	0.175824	1.712155
**In-In**									
0.775299	8.540000	0.567841	2.770000	2.286299	0.088789	1.063382	0.486561	0.461374	1.146285
**P-P**									
4.530000	3.555819	1.804033	1.910011	0.770884	0.450794	1.812700	0.730891	0.473916	1.607665
$\beta_{2}=1$, m=1, $R_{In-P}=3.3$ Å, $R_{In-In}=3.5$ Å, $R_{P-P}=3.0$ Å, D=0.1 Å									

## Data Availability

Not applicable.

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
