# Peer review of "Interatomic Potential for InP"

_materials, 2022, doi:10.3390/ma15144960_

Round 1

Reviewer 1 Report

This paper by Chrobak et al. talks about the development of empirical potentials based on classical MD for large-scale computer simulations. In this work, the authors have adopted a new interatomic potential based on BO model of multibody interactions, along with classical MD simulation, to study B3, B1 phase transition in InP. The MD simulations have been performed using the BO and Branicio potentials. The authors have satisfactorily shown that the crystal modeled by the BO potential shows reversible B1, B3 process (which matches with the experimental observation with X-ray diffraction and Raman spectroscopic studies), whereas that of Branicio potential does not; demonstrating how important it is to choose the correct potential to accurately model elastic properties of materials. I therefore wholeheartedly support its publication in this journal.

 One minor point: The authors should comment on why Branicio potential gives the value of transformation pressure very close to experimental results, as compare to that of BO potential. Also, BO potential models lower transition pressure which is in contrast to the result from enthalpy analysis. A qualitative explanation of these observations will be helpful for the readers.

In line 26, the word ‘phse’ should be replaced by “phase”.

Author Response

.

Reviewer 2 Report

1. The conclusions were generally not sufficiently comprehensive and not innovative.

2.  please cite recent references related to the subject.

3. Do all  the changes highlighted in the attached paper    

Author Response

.

Reviewer 3 Report

The authors present another interatomic potential in order to describe the semiconductor InP. They used MD simulation to describe a specific situation – the structural phase transition, and its reversibility, induced by pressure. The analytical bond order potential proposed was compared with first-principle calculation. That is fine if no experimental results are available.

Nonetheless, I have some questions to be considered.

1)      In equations (1) to (6) there are several quantities not defined, for instance, R, D, etc.

2)      In line 69 it is written, “ …erability of the analytical bond-order (BO) potential, the future which was previously…”, instead future, I guess it is feature.

3)      In line 175-176, it is written “Other atoms behave similarly, and finally, they occupy positions appropriate for the B3 phase unit cell.”, should be B1 phase?

4)      References in Table 1 should be given in full (authors names, journal, etc) in the REFERENCES section.

5)      The reference given in Table 1 – J. Molecular Structure (1992)260, could not be found. Please check if it is correct.

Besides these simple questions, I have another concern. That is ok to compare your simulation with first-principle one if no experimental results exist. For instance, the comparison of the cohesive energy and energy per bond can be found experimentally, among others. In this way the comparison with Branco’s results will be more reliable.

I count 36 parameters needed for the bond-order potential. 10 for each pair of atoms (In-P, In-In and P-P) and other 6 parameters.

 Which are the physical properties used to fit these parameters?

 Since the parameters in the interatomic potential uses several experimental/DFT data to calibrate, it is natural to have a good agreement with the values used, as shown in Table 1. This agreement is not fortuitus.

Beside the structural phase transition induced by pressure, could the bond-order potential describe thermodynamics properties such as melting, specific heat, Debye temperature, phonon vibrational density of states etc? These are some thermodynamics data which, I suppose) are not used in the parameterization of the potential. As stated by Brenner (Phys. Stat. Sol. (b) 217, 23 (2000)), a good interatomic potential is that one which can describe other properties not used in the parameterization.

Author Response

.

Reviewer 4 Report

In this manuscript, the authors present the development of the analytical bond-order potential designed for InP. They study the B3 B1 phase transition in InP using the MD method and a new interatomic potential based on the bond-order model of multi-body interactions. Their potential also reproduces the formation energy of various native point defects in the B3 phase under In-rich and P-rich conditions. They also confirm that the B3 B1 phase transformation employs the orthorhombic mechanism and its reversible nature by applying BO potential for the phase transformation in InP model. This is interesting and informative by itself and the analysis would be helpful for understanding and developing new fast sintering methods for ceramics. I recommend its publication after addressing below minor issues.

1.The abstract should be modified. The author should describe the reason for studying InP and the necessity of developing a new potential in the abstract.

2. It would be better to describe B3 and B1 phases in the introduction. And the original of the B3 B1 phase transition in InP, and the effects of such phase transition.

3. In Table 2, D0, S, , and R0, n, , c, d, h, λ3 need to be explained in the table caption.

4. In line 156, “The panel (a) presents a 20 Åthick…”,the author should add a space between Å and thick.

5. The proposed interaction model describes the reversibility of B3 B1 phase transition. The details of the B1 B3 phase transformation need to be added in the manuscript or SI.

Author Response

.
